# Robotic Stereotactic Body Radiation Therapy for Oligometastatic Liver Metastases: A Systematic Review of the Literature and Evidence Quality Assessment

**DOI:** 10.3390/diagnostics14101055

**Published:** 2024-05-19

**Authors:** Ilektra Kyrochristou, Ilias Giannakodimos, Maria Tolia, Ioannis Georgakopoulos, Nikolaos Pararas, Francesk Mulita, Nikolaos Machairas, Dimitrios Schizas

**Affiliations:** 1Second Department of Surgery, General Hospital of Nikaia, 18454 Athens, Greece; electra.cyro@gmail.com (I.K.); iliasgiannakodimos@gmail.com (I.G.); 2Department of Radiation Oncology, School of Medicine, University of Crete, 71300 Heraklion, Greece; mariatolia1@gmail.com; 3Radiation Oncology Unit, First Department of Radiology, Medical School, Aretaieion Hospital, National and Kapodistrian University of Athens, 11527 Athens, Greece; georgakopoulosioannis@gmail.com; 4Third Department of Surgery, Attikon University Hospital, National and Kapodistrian University of Athens, 12462 Athens, Greece; npararas@gmail.com; 5Department of General Surgery, University General Hospital of Patras, 26504 Patras, Greece; oknarfmulita@hotmail.com; 6Second Department of Propaedeutic Surgery, Laikon General Hospital, National and Kapodistrian University of Athens, 11527 Athens, Greece; nmachair@gmail.com; 7First Department of Surgery, Laikon General Hospital, National and Kapodistrian University of Athens, 11527 Athens, Greece

**Keywords:** robotic stereotactic body radiation therapy, liver metastases, oligometastatic disease, robotics, treatment plan evaluation, imaged radiotherapy

## Abstract

Introduction: The role of stereotactic body radiation therapy (SBRT) as a locally effective therapeutic approach for liver oligometastases from tumors of various origin is well established. We investigated the role of robotic SBRT (rSBRT) treatment on oligometastatic patients with liver lesions. Material and Methods: This review was conducted according to the Preferred Reporting Items for Systematic Reviews and Meta-Analyses statement. The PubMed and Scopus databases were accessed by two independent investigators concerning robotic rSBRT for liver metastases, up to 3 October 2023. Results: In total, 15 studies, including 646 patients with 847 lesions that underwent rSBRT, were included in our systematic review. Complete response (CR) after rSBRT was achieved in 40.5% (95% CI, 36.66–44.46%), partial response (PR) in 19.01% (95% CI, 16.07–22.33%), whereas stable disease (SD) was recorded in 14.38% (95% CI, 11.8–17.41%) and progressive disease (PD) in 13.22% (95% CI, 10.74–16.17%) of patients. Progression-free survival (PFS) rates at 12 and 24 months were estimated at 61.49% (95% CI, 57.01–65.78%) and 32.55% (95% CI, 28.47–36.92%), respectively, while the overall survival (OS) rates at 12 and 24 months were estimated at 58.59% (95% CI, 53.67–63.33%) and 44.19% (95% CI, 39.38–49.12%), respectively. Grade 1 toxicity was reported in 13.81% (95% CI, 11.01–17.18%), Grade 2 toxicity in 5.57% (95% CI, 3.82–8.01%), and Grade 3 toxicity in 2.27% (955 CI, 1.22–4.07%) of included patients. Conclusions: rSBRT represents a promising method achieving local control with minimal toxicity in a significant proportion of patients. Further studies are needed to evaluate the role of rSBRT in the management of metastatic liver lesions.

## 1. Introduction

The liver is a frequent site of cancer metastasis primarily from gastrointestinal, lung, and breast malignancies and represents a common site of cancer recurrence and failure of surgical treatment [1,2]. Advances in understanding intrahepatic anatomy and surgical techniques has led to the performance of major hepatic resection with acceptable morbidity and mortality rates [3]. Although hepatic resection is considered the optimal management liver metastases, only 10–20% of patients are suitable candidates for surgery [4]. Due to technical challenges associated with the surgical treatment of liver metastasis, several alternative therapeutic modalities have been described in the literature, including radiofrequency ablation (RFA), trans-arterial chemoembolization (TACE), laser-induced thermotherapy (LITT), selective internal radiation therapy (SIRT), and stereotactic body radiation therapy. Interestingly, the role of SBRT as a cytoreductive approach for liver oligometastases is well established, demonstrating effective local control and favorable overall survival rates in oligometastatic patients [5,6,7].

The aim of this study was to systematically review the literature regarding the role of rSBRT in the management of metastatic liver disease from tumors of various origins, with a focus on evaluating clinical outcomes and treatment-related toxicity in treated patients. To our knowledge, this is the first systematic review that evaluates the role of rSBRT in the management of liver metastasis.

## 2. Material and Methods

### 2.1. Literature Search and Inclusion Criteria

This systematic review was conducted in accordance with the Preferred Reporting Items for Systematic Reviews and Meta-analyses (PRISMA) guidelines [8]. Two independent investigators (EC and IG) searched the PubMed/Medline, Scopus and Google Scholar databases for articles related to rSBRT (last search: 3 October 2023). Boolean operators (AND, OR) in combination with the following keywords were utilized: “robotic radiotherapy AND liver”, “robotic radiotherapy AND liver metastasis”, “robotic AND Stereotactic Body Radiation Therapy”.

Two investigators (IK and IG), worked independently, searched the databases, screened the articles retrieved, and further assessed the eligibility of articles in an abstract form or in full text, if accessible. Each author made decisions for article inclusions based on the following predetermined eligibility criteria. A language restriction was applied, and articles written only in English were considered suitable for this systematic review. Randomized controlled trials and retrospective and prospective studies that reported on the use of robotic SBRT were included. On the contrary, articles that reported on non-robotic SBRT were excluded. Additionally, case reports, letters to the editor, comments, reviews, systematic reviews, and animal surveys were excluded. All the references of the eligible articles were checked in order to identify potential abstracts through the snowball procedure. The selection process is summarized in a flow-chart (Figure 1).

According to the RECIST criteria, in the majority of included studies, the complete response (CR) rate was defined as the elimination of all target lesions, partial response (PR) rate as at least a 30% decrease in the diameter of target lesions, and progressive disease (PD) as at least a 20% increase in the diameter of target lesions, while no change in the size of target lesions was classified as stable disease (SD) [9]. Furthermore, according to the ASTRO and ESTRO consensus documents, in the majority of published studies, oligometastatic disease was defined as 1–5 metastatic lesions to the liver, with all metastatic sites to be treatable, with the controlled status of the primary tumor being optional [10].

### 2.2. Data Extraction

Data extraction was performed by two independent researchers (EC and IG) using a pre-defined template. The following variables were collected and tabulated: study characteristics (sample size, publishing date, type of study), number of patients, number of lesions, administered dose, imaging technique for lesion confirmation, response rates, local or distant progression of tumor, patient survival, and mortality. Grade of toxicity and predictive factors were also recorded.

At each stage, reasons for excluding data were documented. Disagreement concerning article selection was resolved by the interreference of a third member of the review team (DS). All studies were saved in an Endnote Database.

### 2.3. Statistical Analyses

For the continuous variable, the final value, the standard deviation (SD) of the outcome and number of patients assessed in each treatment arm were extracted. A pooled estimate of treatment effect was calculated by the mean difference (MD) and its 95% confidence interval (95% CI). Statistical analysis was carried out using IBM SPSS Statistics for Windows, Version 24.0. Armonk, NY: IBM Corp. All outcomes were qualitatively summarized.

### 2.4. Assessment of Study Quality

The quality of included case series was assessed using the tool developed by the National Heart, Lung, and Blood Institute (NHLBI) based on work from the Agency for Healthcare Research and Quality, the Cochrane Collaboration, the United States Preventive Services Task Force, the Scottish Intercollegiate Guidelines Network and the National Health Service Centre for Reviews and Dissemination. The NHLBI scale ranges from 1 to 9, with a score of 1–3 demonstrating poor quality, 4–6 fair quality, and 7–9 showing good quality. The mean and SD values for the NHLBI score of this systematic review were 7.6 (mean) +− 0.61 (SD). Three independent reviewers (IG, IC, DS) rated the quality of included studies, and a synthesis of their reports was performed (Appendix A).

## 3. Results

### 3.1. Article Selection and Studies Characteristics

The database search retrieved 1018 records and resulted in 87 unique articles after duplicate removal. After the completion of the selection process, only 15 studies fulfilled the inclusion criteria and were analyzed [11,12,13,14,15,16,17,18,19,20,21,22,23,24,25]. Overall, 646 patients with 847 lesions that underwent rSBRT were included in this systematic review and analyzed. Out of the available data, doses ranged from 9 to 45 Gy. New or remaining lesions after the employment of rSBRT were identified by using enhanced CT and MRI in seven studies (46.7%) [12,13,15,17,20,22,23], only enhanced CT in three studies (20%) [18,21,25], 18FDG PΕT/CT along with MRI in four studies (26.7%) [11,14,16,19] and enhanced MRI alone in one study (6.7%) [24]. The mean follow-up of included studies was estimated at 17.9 +− 5.44 (SD) months and ranged from 13 to 30 months. The characteristics of included studies are summarized in Table 1.

### 3.2. Location of Primary Tumor

Out of the included patients with liver metastasis, the majority of primary tumors was located in the colon or rectum (315 patients, 48.76%, 95% CI, 44.93–52.61%). Primary tumors also originated from the breast in 55 patients (8.51%, 95% CI, 6.59–10.93%), urogenital tract in 29 patients (4.49%, 95% CI, 3.12–6.39%), lung in 31 patients (4.8%, 95% CI, 3.38–6.75%), gastrointestinal tract in 36 patients (5.57%, 95% CI, 4.03–7.64%), and liver in 9 patients (1.39%, 95% CI, 0.69–2.67%). In 94 patients (14.55%, 95% CI, 12.03–17.49%), the primary tumor was found in other organs, such as pancreas, gallbladder, or in unspecified primary locations. The primary location of tumors with metastasis to the liver is summarized in Table 2.

### 3.3. Short-Term Outcomes

In total, eleven studies [11,12,13,15,16,17,18,20,22,23,25], including 485 patients, evaluated the side effects of rSBRT administration in treated patients. Out of the available data, Grade 1 acute toxicity was reported in 67 patients (13.81%, 95% CI, 11.01–17.18%), Grade 2 acute toxicity in 27 patients (5.57%, 95% CI, 3.82–8.01%), and Grade 3 acute toxicity in 11 patients (2.27%, 95% CI, 1.22–4.07%). Detailed data concerning post-radiation side effects are described in Table 3.

### 3.4. Response Rate after rSBRT

In total, 10 studies [11,12,13,14,16,22,23,24,25], including 605 metastatic lesions to the liver, recorded response rates after the administration of rSBRT. Out of the available data, CR rates ranged from 0 to 65.2%, PR rates from 13.1 to 64.3%, while SD varied between 8% and 39% and PD between 5.5% and 44.8%. Out of the available data, after the rSBRT delivery, CR was achieved in 245 treated lesions (40.5%, 95% CI, 36.66–44.46%) and PR in 115 treated lesions (19.01%, 95% CI, 16.07–22.33%), whereas SD was recorded in 87 treated lesions (14.38%, 95% CI, 11.8–17.41%) and PD in 80 treated lesions (13.22%, 95% CI, 10.74–16.17%). The response rates and status of disease after the rSBRT delivery are summarized in Appendix A.

### 3.5. Long-Term Outcomes

#### 3.5.1. Progression-Free Survival

In total, ten studies [10,11,12,17,19,20,21,22,23] presented data concerning the PFS rates and median-free survival after rSBRT. Out of the available data, four studies [11,20,21,22,23] evaluated the role of rSBRT in patients with metastatic liver disease of colorectal origin only. Vernaleone et al. [11] found that the median PFS was 6.55 months, while the PFS rates in one year and two years of follow-up were estimated at 17.6% and 11.6%, respectively. Furthermore, PFS rates were estimated at 73.3% in one year and at 67.4% in two years in a study conducted by Dewas et al. [21], while they were estimated at 85% in one year and 80% in two years by Strintzing et al. [23]. Finally, in another study, Strintzing et al. found a median free survival period of 9.2 months [20].

Additionally, six studies [10,12,13,17,19,22] evaluated the progression-free rates in patients with metastases to the liver derived from various primary origins, as seen in Table 2. Andratchke et al. [17] found that the free survival rates after one and two years were estimated at 35.1% and 17.7%, respectively, in patients with colorectal, breast, lung, and other primary tumors, and Strintzing et al. found them at 85% and 80%, respectively, in patients with colorectal tumor origin [20]. Better progression-free survival rates after one and two years of follow-up were reported by Vautravers et al. [22] at 90% and 86%, respectively, in patients with primary tumors of various origin, while Anstadt et al. [12] reported lower progression-free rates in patients with tumors of different origin. Finally, Yuan et al. [19] reported median progression-free survival at 12 months in patients with tumors originated from various organs, such as the lungs, breasts, and colorectal and urogenital systems. PFS rates after rSBRT are summarized in Table 4.

#### 3.5.2. Overall Survival

In addition, 11 studies [11,12,13,14,19,20,21,22,23,24] collected data regarding the overall survival of patients after the rSBRT delivery. Interestingly, only two studies recorded the overall survival rates of rSBRT in colorectal cancer cases. Vernaleone et al. [11] reported 66.2% and 49.6% overall survival at one year and two years, respectively, while the median overall survival was estimated at 20.1 months. Furthermore, Strintzing et al. reported the median overall survival at 34.4 months [24].

In total, nine studies [12,13,14,15,19,20,21,22,23] evaluated the overall survival of SBRT in patients with cancers of various origin. Berkovic et al. [15] reported favorable overall survival rates in the one- and two-year follow-ups, estimated at 87.2% and 78.3%, respectively, in patients with colorectal, lung, stomach, and breast origin of primary tumors. The overall survival rates of the included studies are shown in Table 4.

### 3.6. Prognostic Indicators of rSBRT

Prognostic factors of rSBRT are summarized in Table 5.

## 4. Discussion

Although SBRT has been extensively studied, the exact role of robotic SBRT (rSBRT) in the management of liver metastases has not been definitely established. rSBRT represents a novel, non-invasive radiotherapeutic approach that allows a highly customizable treatment planning, leveraging real-time imaging and robotic precision to deliver radiation with sub-millimeter accuracy. This technology utilizes the CyberKnife image-guided stereotactic radio-surgical system enabling a real-time tumor tracking and surgeons to apply high radiation doses within the gross tumor volume (GTV) and achieve increased protection of the surrounding healthy tissue [7].

The optimal therapeutic approach capable of ensuring extended disease control or complete remission in patients with liver metastasis is the surgical excision of the metastatic lesion, with 5-year survival rates projected between 40 and 55% [26]. However, as already mentioned, only a minority of patients meet the criteria for partial hepatectomy, thus prompting the development of alternative thermal and non-thermal ablative therapies for managing unresectable liver metastases [26]. Interestingly, thermal ablation seems to be effective and safe in treating colorectal liver metastatic lesions <3 cm but exhibits increased rates of local tumor progression in larger lesions (>3 cm) [27]. More specifically, findings from the EORTC-CLOCC trial revealed the improved overall survival (OS) of radiofrequency ablation (RFA) combined with chemotherapy compared to chemotherapy alone (HR = 0.58; 95% CI: 0.38–0.88) [28]. Additionally, in a randomized phase II trial, patients with unresectable colorectal liver metastases received either systemic treatment alone or systemic treatment plus aggressive local treatment by RFA or resection [28]. At a 9.7-year median follow-up, the mortality rates were 65% and 89.8% in the combined treatment and systemic treatment alone groups, respectively, with a statistically significant difference in OS (HR: 0.58, 95% CI: 0.38 to 0.88) [28]. MWA systems are considered superior alternatives to conventional RFA for larger lesions [29,30]. Notably, in a systematic review, local tumor progression rates ranged from 11 to 78% for RFA and 14% to 38% for MWA, suggesting a preference for lesions >3 cm for MWA [26]. Furthermore, another local treatment option constitutes SBRT, which exhibits greater efficacy for larger lesions and challenging anatomical locations [26]. In a retrospective cohort by Jackson et al., SBRT demonstrated a superior freedom from local progression compared to RFA, especially for hepatic metastases > 2 cm, while no difference in median OS was found [31], indicating better local control with SBRT than thermal ablation for larger lesions. Finally, TACE comprises another treatment option, especially in patients who have failed systematic therapy, aiming to achieve tumor control mainly for palliative purposes, presenting with high tumor regression and disease control rates [32].

Due to liver’s low tolerance to irradiation, the focus of local treatment planning has shifted towards methods that deliver conformal radiation doses to tumors while minimizing exposure to critical surrounding tissues [33,34]. rSBRT offers an advantage in this regard, delivering high doses of radiation in one or few fractions, with precise targeting, high accuracy, and sharp fall-off dose to spare healthy hepatic tissue. Interestingly, the effectiveness of rSBRT allows physicians to treat oligometastases with a single dose [23]. Overall, different regimens of 24–60 Gy in one or few fractions are considered comparable to other forms of local therapy for the management of unresectable liver metastases [23]. In our systematic analyses, the delivered rSBRT doses ranged from 9 to 45 Gy.

Concerning the number of fractions delivered, Stintzing et al. utilized a single fraction of 24 Gy or higher, peaking at 45 Gy during rSBRT sessions. Of note, they reported significantly higher complete and partial response rates, ranging from 64.3% to 78.6%, compared to repetitive irradiation treatments. In our systematic analysis, CR was achieved in 41.53% (95% CI, 36.91–46.3%) and PR in 20.76% (95% CI, 17.15–24.91%) of included patients. Interestingly, higher radiation doses per fraction correlate with increased tumor response, supporting a dose–response relationship [35]. The majority of researchers tend to favor a repetitive radiation plan, consisting of 3 to 5 doses ranging from 9 to 45 Gy, aiming to keep the progressive disease below 30%. Multiple sessions of low dose energy delivery are preferred based on data showing that one-fraction therapies are less effective than multi-dose treatments with systems other than CyberKnife [36].

Concerning rSBRT efficacy, median PFS ranged from 6.5 to 34.4 months in our systematic review. Regarding liver metastases of colorectal origin, Dewas et al. found [21] 1-year progression-free survival rates at 73.3% and 2-year progression-free survival rates at 67.4%. However, Vernaleone observed that progression-free survival after one year and two years of follow-up were estimated at 17.6% and 11.6%, respectively [11]. Additionally, concerning overall survival following rSBRT in patients with cancers of various origin, Berkovic reported favorable one- and two-year OS rates, estimated at 87.2% and 78.3%, respectively [20]. Although Vautravens et al. [22] recorded better 1-year overall survival (94%) in patients with primary tumors of various origin, 2-year overall survival diminished significantly (48%). Of note, Dutta et al. [14] recorded the lowest survival rates in the one- and two-year follow-ups of patients with tumors of colorectal and breast origin, at 38% and 5%, respectively. In the remaining studies, the overall survival rates in 1-year and 2-year follow-ups ranged from 72% to 89.6% and from 62% to 72.2%, respectively. Interestingly, out of the available data, Strintzing et al. [13] reported higher median survival after r-SBRT at 35.2 months in patients with tumors of various origin. Of note, differences in reported survival rates may depend on the biological tumor behavior and experience of radiologists in diagnosing remaining lesions after radiation. In the majority of published studies, deaths following SBRT were mainly attributed to tumor progression. Andratschke et al. reported that nearly 90% of deaths resulted from hepatic or extrahepatic disease progression, while Vautravers-Dewas et al. recorded a 23.8% mortality rate due to extrahepatic disease progression [17,22].

In total, patients treated with radiotherapy using the CyberKnife system present increased tolerability, attributed to enhanced therapeutic accuracy and reduced toxicity to surrounding tissues. Of note, the pathophysiological mechanisms of radiation-induced normal tissue damage are similar for biologically equivalent single and fractionated dose of irradiation [37,38]. In our systematic analysis, Grade 1 toxicity was reported in 24.5% (95% CI, 20.29–29.27%), Grade 2 toxicity in 5.13% (95% CI, 3.22–8.01%), and Grade 3 toxicity in 2.28% (95% CI, 1.08–4.51%) of included patients. Interestingly, 80% (12 studies) of included studies used Common Terminology Criteria for Adverse Events (CTCAE) criteria for grading toxicity, while the WHO criteria and RTOG criteria were used only in one study, respectively. CTCAE criteria have now been adopted for grading toxicity in the majority of cancer studies and clinical trials since their percentage agreement with the patients’ own experiences is considerably better than that of the WHO score [39]. However, in our systematic review, the majority of included studies used the CTCAE criteria; thus, no great variance in toxicity rates can be attributed to different grading score systems. Although several factors seem to affect the efficacy of rSBRT in the management of metastatic liver lesions, these factors seem to differ between published studies. Among the included studies that underwent a univariate analysis, age, previous extrahepatic disease, the number and size of hepatic lesions and the histology of the primary tumor constitute significant prognostic indicators of rSBRT [11,12,13,14,15,17,18,21]. However, only previous extrahepatic disease, tumor volume, and colorectal origin of primary tumor were found as poor independent indicators in the multivariate analysis of the included studies [11,12,13,14,15,17,18,21]. The majority of researchers agree that the colorectal origin of the primary tumor represents a negative prognostic factor for these patients. Furthermore, Vernaleone et al. suggested that rSBRT may find its highest efficacy in tumors with total volume between 3 and 5 cm [11].

Robotic radiosurgery offers the advantage of real-time tumor tracking, enabling the application of high radiation doses in a single treatment session [20,40]. Interestingly, the CyberKnife system can detect and respond to patient’s respiratory movements and, thus, should be preferred for elderly patients who are unable apply breath-holding techniques during RFA procedures. Limited studies compared robotic SBRT with non-invasive local treatments for metastatic liver tumors. Strintzing et al. aimed to assess the treatment efficacy and toxicity of RFA versus robotic radiosurgery in patients with colorectal liver metastases [20]. Robotic radiosurgery presented with higher, but non-significant, one-year and two-year local control rates compared to RFA-treated patients, while the RFA group exhibited a longer median overall survival (*p* = 0.06) [20].

In patients with synchronous colorectal liver metastases, the surgical strategy, including primary-first, liver-first, or simultaneous resections, should be decided according to the hepatic tumor burden [41]. More specifically, the three approaches present with equivalent results for low-complexity surgical procedures, while the simultaneous resection group has worse outcomes for major hepatectomies and resections of multiple liver metastases [41]. Of note, the expert group on oncosurgery management of liver metastases stated that simultaneous resection, when feasible without increasing operative risk, is the preferred option [42]. However, in these patients, the majority of liver metastases are initially deemed unresectable and require neoadjuvant chemotherapy to potentially render the metastases resectable [42]. Interestingly, there are several conditions for an optimal oncosurgical approach. The first is optimal first-line chemotherapy because there is a strong correlation between the resection rate and the response rate to chemotherapy [43]. The second condition is a short duration of first-line chemotherapy since conducting more cycles before surgery may increase liver toxicity and subsequently limit the number of cycles feasible after surgery [42]. Overall, the operative risks of simultaneous resections depend on complexity of both hepatectomy and colorectal resections and the increased mortality risk of the simultaneous approach in complex procedures is a major point, while, in the presence of multiple bilobar liver metastases, the liver-first approach is associated with longer survival rates than the alternative approaches and should be evaluated as the standard [41].

To our knowledge, this is the first systematic review in the literature concerning the evaluation of R-SBRT for the therapeutic management of patients with liver metastases. The methodological strengths of the present paper include the following: (1) comprehensive literature search following systematic methodology, (2) accurate data extraction with pre-defined templates, and (3) standardized quality assessment of eligible studies using the NHLBI quality assessment tool. However, this systematic analysis has certain limitations. As with any systematic review, the included articles did not record all outcomes of interest; therefore, relative rates and percentages were estimated based on available data. Furthermore, the vast majority of included studies concern retrospective studies from single institutions presented with insufficient data and their credibility mainly depends on accurate record keeping. The heterogeneity among institutions concerning primary origin of cancer with metastasis to the liver, records of short-term and long-term outcomes and patient follow-up might affect survival outcomes and time-to-event analysis. More specifically, due to the heterogeneity of primary tumor origin among studies, further analysis of long-term progression-free and overall survival rates was not possible.

## 5. Conclusions

Robotic SBRT represents an effective alternative treatment for the management of patients with inoperative metastatic tumors to the liver. Patients undergoing rSBRT demonstrate favorable overall survival and progression-free survival rates, along with low incidence of short-term toxicity, compared to other local treatment modalities. Further well-designed prospective and retrospective studies are required to provide valuable insights into the optimal use and outcomes of robotic SBRT in this clinical setting.

## Figures and Tables

**Figure 1 diagnostics-14-01055-f001:**
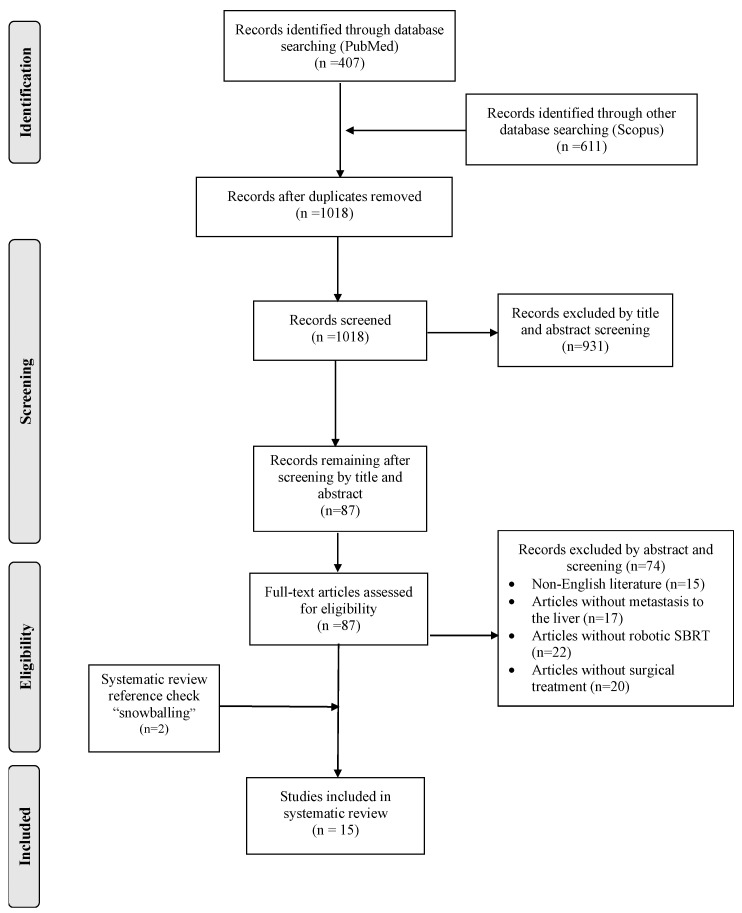
Prisma flow-chart of this study.

**Table 1 diagnostics-14-01055-t001:** Overview of included studies with robotic SBRT.

Author	Article Type	Year of Publication	Number of Patients	Number of Lesions	Number of Fractions/ Dose (Gy)	New Lesions Confirmation	Follow-Up (Months)
Vernaleone et al. [11]	Prospective	2019	38	63	(3–5) × (25–45)	18FDG-CT/PET scan and/or a liver MRI	11.8
Anstadt et al. [12]	Prospective	2019	42	81	3 × 18	enhanced CT scan and liver MRI	25
Stintzing et al. [13]	Prospective	2019	126	194	1 × (20–45)	enhanced CT scan and liver MRI	30
Dutta et al. [14]	Retrospective	2018	18	33	(21–45) × 3	18FDG-CT/PET scan and/or a liver MRI	12.5
Berkovic et al. [15]	Prospective	2017	42	55	(3–6) × 14	enhanced CT scan and liver MRI	18.9
Garcia et al. [16]	Prospective	2016	9	17	(3–5) × 14	18FDG-CT/PET scan and/or a liver MRI	15.2
Andratschke et al. [17]	Retrospective	2016	52	91	3 × 18	enhanced CT scan and liver MRI	17
Janoray et al. [18]	Retrospective	2014	41	-	3 × 15 or 20	enchanced CT scan	14
Yuan et al. [19]	Retrospective	2014	57	80	(21–54) × 3	18FDG-CT/PET scan and/or a liver MRI	20.5
Stintzing et al. [20]	Prospective	2013	30	35	1 × (24–26)	enhanced CT scan and liver MRI	23.3
Dewas et al. [21]	Retrospective	2012	72		4 × 10 and afterwards 3 × 15	enhanced CT scan	14.3
Vautravers-Dewas et al. [22]	Retrospective	2011	42	62	3 × 15 or 4 × 10	enhanced CT scan and liver MRI	14.3
Stintzing et al. [23]	Prospective	2010	36	54	1 × 24	enhanced CT scan and liver MRI	21.3
Stintzing et al. [24]	Prospective	2010	14	19	1 × 24	enhanced MRI	16.8
Ambrosino et al. [25]	Prospective	2009	27	63	3 × (9–20)	enhanced CT scan	13

Gy, Gray; -, Non reported.

**Table 2 diagnostics-14-01055-t002:** Primary location of tumors of patients included in this systematic review.

Authors	Number of Patients	Location n, % (95% CI)						
		Colorectal	Gastrointestinal	Breast	Uro-Genital	Lungs	Liver	Other
Vernaleone et al. [11]	38	38	-	-	-	-	-	-
Anstadt et al. [12]	42	18	6	3	5	7	-	3
Stintzing et al. [13]	126	71	13	14	15	5	1	8
Dutta et al. [14]	18	13	-	5	-	-	-	-
Berkovic et al. [15]	42	30	2	11	-	3	-	8
Garcia et al. [16]	9	1	1	4	2	-	-	1
Andratschke et al. [17]	52	22	-	4	-	4	-	22
Janoray et al. [18]	41	-	-	-	-		-	-
Yuan et al. [19]	57	18	5	7	2	7	5	13
Stintzing et al. [20]	30	30	-	-	-		-	-
Dewas et al. [21]	72		-		-		-	-
Vautravers-Dewas et al. [22]	42	30	3	3	-	3	-	6
Stintzing et al. [23]	36	19	5	2	4	1	3	2
Stintzing et al. [24]	14	14	-	-	-		-	-
Ambrosino et al. [25]	27	11	1	2	1	1	-	11
**Total**	**646**	**315, 48.76% (44.93–52.61%)**	**36, 5.57% (4.03–7.64%)**	**55, 8.51% ** **(6.59–10.93%)**	**29, 4.49% (3.12–6.39%)**	**31, 4.8% (3.38–6.75%)**	**9, 1.39% (0.69–2.67%)**	**94, 14.55% (12.03–17.49%)**

CI, Confidence Interval.

**Table 3 diagnostics-14-01055-t003:** rSBRT induced toxicity.

Authors	Number of Patients	Toxicity		
		Grade 1 Pts (%)	Grade 2 Pts (%)	Grade 3 Pts (%)
Vernaleone et al. [11]	38	4 (10.5%)	3 (7.9%)	0
Anstadt et al. [12]	42	12 (28%)	2 (7%)	0
Stintzing et al. [13]	126	1 (0.79%)	2 (1.59%)	2 (1.59%)
Berkovic et al. [15]	42	0	6 (11%)	3 (5%)
Garcia et al. [16]	9	3 (33.3%)	0	0
Andratschke et al. [17]	52	12 (24.1%)	1 (1.9%)	0
Janoray et al. [18]	41	0	0	2 (5.71%)
Stintzing et al. [20]	30	3 (10%)	1 (3%)	0
Vautravers-Dewas et al. [22]	42	19 (45.2%)	3 (7.14%)	1 (3.4%)
Stintzing et al. [23]	36	13 (36%)	0	0
Ambrosino et al. [25]	27	0	9 (33.3%)	3 (11.1%)
**Total, %** **(95% CI)**	**485**	**67, 13.81%** **(11.01–17.18%)**	**27, 5.57%** **(3.82–8.01%)**	**11, 2.27%** **(1.22–4.07%)**

**Table 4 diagnostics-14-01055-t004:** Progression-free and overall survival of included patients after rSBRT.

Authors	Progression-Free Survival			Overall Survival		
	Median (Months)	1 Year (%)	2 Years (%)	Median (Months)	1 Year (%)	2 Years (%)
Vernaleone et al. [11]	6.55	17.6%	11.7%	20.1	66.2%	49.6%
Anstadt et al. [12]	10	32%	23%	30	72%	62%
Stintzing et al. [13]	11.9	50%	35.2%	35.2	80%	70%
Dutta et al. [14]	-	-	-	6.5	38%	5%
Berkovic et al. [15]	-	55%	42.3%	-	87.2%	78.3%
Andratschke et al. [17]	-	35.1%	17.7%	23	70.2%	45%
Yuan et al. [19]	12	-	-	37.5	89.6%	72.2%
Stintzing et al. [20]	34.4	85%	80%	34.4	-	-
Dewas et al. [21]	8	73.3%	-	-	-	-
Vautravers-Dewas et al. [22]	6.5	90%	86%	-	94%	48%
Stintzing et al. [23]	11.6	83%	25.1%	25.1	83%	70%
Stintzing et al. [24]	9.2	-	-	-	-	-
**Total %** **(95% CI)**		**61.49%** **(57.01–65.78%)**	**32.55%** **(28.47–36.92%)**		**66.41%** **(62.27–70.32%)**	**52.18%** **(47.92–56.42%)**

**Table 5 diagnostics-14-01055-t005:** Poor prognostic indicators of rSBRT reported in the literature.

Authors	Univariate Analysis	Multivariate Analysis
Vernaleone et al. [11]	age (*p* = 0.045) previous extrahepatic disease (*p* = 0.0001) number of metastatic lesions > 3 before the treatment (*p* = 0.049)	previous extrahepatic disease (*p* = 0.0001)
Stritizing et al. [13]	lesions > 5 cm in diameter (*p* = 0.032)	NM
Dutta et al. [14]	primary tumor size (*p* = 0.0001)	NM
Berkovic et al. [15]	performance status (0 vs. 1+) histology in colorectal cancer cases (adenocarcinoma vs. other)	NM
Andratschke et al. [17]	KPS (*p* < 0.05) histology (breast and colorectal cancer having a better prognosis) GTV volume	tumor volume (*p* < 0.05) colorectal cancer (*p* < 0.05)
Janoray et al. [18]	hepatic lesion at least 35 mm	NM
Dewas et al. [21]	PTV > 200 cc (*p* = 0.014)	NM

GTV, Gross Tumor Volume; NM, Not mentioned; KPS, Karnofsky Performance status; PTV, Pathologic Tumor Volume.

## Data Availability

The original data presented in the study are openly available https://doi.org/10.5281/zenodo.11215122.

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
