# Peer review of "Robotic Stereotactic Body Radiation Therapy for Oligometastatic Liver Metastases: A Systematic Review of the Literature and Evidence Quality Assessment"

_diagnostics, 2024, doi:10.3390/diagnostics14101055_

Round 1
Reviewer 1 Report
Comments and Suggestions for Authors
This is thorough analysis by PRISMA criteria on a narrow (due to the robotic component), but important topic (the liver mets).
A clear statement on the INCLUSION CRITERIA is needed , and shall include the OLIGOMETS status (as stated in the abstract, and the term OLIGOMETASTATIC maybe deserves the inclusion in the title), checked in each trial- explaining maybe the huge differences between selected articles as to PFS & OS, as the ESTRO consensus paper was published only in 2020/ Lancet Oncol.....);
- definition on the toxicity scale used on different trials (CTCAE vs RTOG....) and discussion accordingly, for the differences among articles;
RESULTS should state only raw data, without discussion them. Therefore, for example rows 190 to 197 or 202 to 208 belong, for me, to the discussions.
Consider revising the false statements below regarding surgeons/ surgery (row 218-221): @tumor tracking, enabling surgeons to apply high radiation doses within the gross tumor volume (GTV) and achieve minimal safety margins along with protection of the surrounding healthy tissue as much as possible (7).
Surgical resection of liver metastasis originating from various primary regions has limited indications,@
Please clarify/reconsider phrasing of (row244...): "Delivered doses are either stable or fluctuating, depending on tumor’s size and behavior along with radiation’s toxicity." AND(row253) "survival rates may vary due to diversity in biological tumor behavior, but also due to the fluctuating experience of radiologists in diagnosing remaining lesions post radiation." AND (row 262 )Of note, the pathophysiological mechanisms of radiation-induced damage to the healthy surrounding tissues do not differ for biologically equivalent doses of irradiation, as proved by phantom dosimetric studies and exper-264 imental studies (29, 30).-
In order to support the CONCLUSION, in depth discussion comparing the RFA/MWA, TACE, non robotic SBRT,SIRT or combo TACE +RT, and surgical series is needed, including the appropiate references (for ex. Ruers T, Van Coevorden F, Punt CJ et al. Results of a Randomized Phase II Trial. J Natl Cancer Inst. 2017 Sep 1;109(9). doi: 10.1093/jnci/djx015.)
Author Response
Athens, 15/05/2024
Dear Reviewer,
Thank you for reviewing our manuscript entitled “Robotic Stereotactic Body Radiation Therapy for oligometastatic liver metastases: a systematic review of the literature and evidence quality assessment.”
We are pleased that the manuscript was favorably reviewed and was found to be potentially acceptable for publication pending revisions. We thank the reviewers for the valuable insight and comments as these serve to further strengthen our manuscript.
As requested, we have provided a point-by-point response to each of the reviewers’ comments with relevant changes made to the manuscript highlighted in yellow. References, where appropriate, are listed within the corresponding section.
Point-by-point reply to reviewers.
REVIEWER 1.
Remark 1. This is thorough analysis by PRISMA criteria on a narrow (due to the robotic component), but important topic (the liver mets).
A clear statement on the INCLUSION CRITERIA is needed, and shall include the OLIGOMETS status (as stated in the abstract, and the term OLIGOMETASTATIC maybe deserves the inclusion in the title), checked in each trial- explaining maybe the huge differences between selected articles as to PFS & OS, as the ESTRO consensus paper was published only in 2020/ Lancet Oncol.....);
Our response. Thank you for your valuable comment. All studies incorporated the term "oligometastatic disease" as part of their inclusion criteria. The majority of these studies defined oligometastatic disease as involving liver lesions ≤4, while one study included patients with <2 liver lesions, and another study did not specify the number of metastatic lesions (although it met the criteria for oligometastatic disease based on the results). As a result, there was no gross heterogeneity in the oligometastatic status of included studies, to explain differences in survival rates among studies. Furthermore, the term “oligometastatic” was added in the Title of the manuscript as you proposed. In addition, the sentence: “Furthermore, according to ASTRO and ESTRO consensus document, in the majority of published studies, oligometastatic disease was defined as 1-5 metastatic lesions to the liver, with all metastatic sites to be treatable, while controlled status of primary tumor being optional (10).” was added in the Material and Methods section.
Remark 2. definition on the toxicity scale used on different trials (CTCAE vs RTOG....) and discussion accordingly, for the differences among articles;
Our response. Thank you for your valuable advice. Toxicity grading systems used on different trials were searched among studies and the following text concerning toxicity grading was added in the Discussion section; “Interestingly, 80% (12 studies) of included studies used Common Terminology Criteria for Adverse Events (CTCAE) criteria for grading toxicity, while WHO criteria and RTOG criteria were used only in one study, respectively. CTCAE criteria have now been adopted for grading toxicity in the majority of cancer studies and clinical trials, since their percentage agreement with the patients' own experiences is considerably better than that of the WHO score (39). However, in our systematic review, the majority of included studies used CTCAE criteria and thus, no great variance in toxicity rates can be attributed to different grading score systems.”
Remark 3. RESULTS should state only raw data, without discussion them. Therefore, for example rows 190 to 197 or 202 to 208 belong, for me, to the discussions.
Our response. Thank you for your constructive suggestion. Rows 190 to 197 and 202 to 208 were removed to the Discussion section as you proposed.
Remark 4. Consider revising the false statements below regarding surgeons/ surgery (row 218-221): @tumor tracking, enabling surgeons to apply high radiation doses within the gross tumor volume (GTV) and achieve minimal safety margins along with protection of the surrounding healthy tissue as much as possible (7).
Our response. Thank you for your valuable remark. The statement you indicated was revised and was replaced by the following sentence: “This technology utilizes the Cyber knife image-guided stereotactic radio-surgical system enabling a real-time tumor tracking and surgeons to apply high radiation doses within the gross tumor volume (GTV) and achieve increased protection of the surrounding healthy tissue (7).”
Remark 5. Surgical resection of liver metastasis originating from various primary regions has limited indications,@
Our response. Thank you for your valuable remark. The following sentence was deleted from the Discussion section.
Remark 6. Please clarify/reconsider phrasing of (row244...): "Delivered doses are either stable or fluctuating, depending on tumor’s size and behavior along with radiation’s toxicity." AND(row253) "survival rates may vary due to diversity in biological tumor behavior, but also due to the fluctuating experience of radiologists in diagnosing remaining lesions post radiation." AND (row 262 ) Of note, the pathophysiological mechanisms of radiation-induced damage to the healthy surrounding tissues do not differ for biologically equivalent doses of irradiation, as proved by phantom dosimetric studies and exper-264 imental studies (29, 30).-
Our response. Thank you for your valuable and detailed remarks. The following sentence “Delivered doses are either stable or fluctuating, depending on tumor’s size and behavior along with radiation’s toxicity." was deleted from the Discussion section. The following sentence "survival rates may vary due to diversity in biological tumor behavior, but also due to the fluctuating experience of radiologists in diagnosing remaining lesions post radiation." was revised and replaced by the following phrase: “Of note, differences in reported survival rates may depend on biological tumor behavior and experience of radiologists in diagnosing remaining lesions after radiation. ”The Following sentence “Of note, the pathophysiological mechanisms of radiation-induced damage to the healthy surrounding tissues do not differ for biologically equivalent doses of irradiation, as proved by phantom dosimetric studies and experimental studies (37,38).” was replaced by the phrase: “Of note, the pathophysiological mechanisms of radiation-induced normal tissue damage are similar for biologically equivalent single and fractionated dose of irradiation (37,38).”
Remark 7. In order to support the CONCLUSION, in depth discussion comparing the RFA/MWA, TACE, non-robotic SBRT,SIRT or combo TACE +RT, and surgical series is needed, including the appropiate references (for ex. Ruers T, Van Coevorden F, Punt CJ et al. Results of a Randomized Phase II Trial. J Natl Cancer Inst. 2017 Sep 1;109(9). doi: 10.1093/jnci/djx015.)
Our response. Thank you for your constructive comment. An extensive discussion comparing different surgical and non-invasive interventions, including the appropriate references, as the one you proposed, were added in the Discussion section: “The optimal therapeutic approach … presenting with high tumor regression and disease control rates.”.
Thank you for considering our revised manuscript.
Reviewer 2 Report
Comments and Suggestions for Authors
Excellent work by colleagues who conducted a systematic search on literature articles to systematically contribute to the approval of robotic SBRT by the scientific community. The possibility of treating secondary liver lesions is currently quite broad, ranging from surgical to stereotactic to TACE to reach the one described in the following paper. This offers the possibility of reaching parts of the liver that would otherwise be difficult to reach, a single treatment is sufficient and at doses calibrated according to the metastatic disease, this offers millimetric precision, with respect for nearby healthy structures. The follow-up offers us the opportunity to examine the long-term results which are, as described, certainly encouraging. I ask the researchers whether a combined treatment of removal of the main lesion together with the secondary hepatic lesion is possible and possibly especially after neoadjuvant or adjuvant treatments, is there an indication to propose it to patients in the same measure. Excellent iconography, English needs improvement. The bibliography supports the initial thesis
Comments on the Quality of English LanguageEnglish needs to be improved
Author Response
Athens, 15/05/2024
Dear Reviewer,
Thank you for reviewing our manuscript entitled “Robotic Stereotactic Body Radiation Therapy for oligometastatic liver metastases: a systematic review of the literature and evidence quality assessment.”
We are pleased that the manuscript was favorably reviewed and was found to be potentially acceptable for publication pending revisions. We thank the reviewers for the valuable insight and comments as these serve to further strengthen our manuscript.
As requested, we have provided a point-by-point response to each of the reviewers’ comments with relevant changes made to the manuscript highlighted in yellow. References, where appropriate, are listed within the corresponding section.
Point-by-point reply to reviewers.
REVIEWER 2
Remark 1. Excellent work by colleagues who conducted a systematic search on literature articles to systematically contribute to the approval of robotic SBRT by the scientific community. The possibility of treating secondary liver lesions is currently quite broad, ranging from surgical to stereotactic to TACE to reach the one described in the following paper. This offers the possibility of reaching parts of the liver that would otherwise be difficult to reach, a single treatment is sufficient and at doses calibrated according to the metastatic disease, this offers millimetric precision, with respect for nearby healthy structures. The follow-up offers us the opportunity to examine the long-term results which are, as described, certainly encouraging. I ask the researchers whether a combined treatment of removal of the main lesion together with the secondary hepatic lesion is possible and possibly especially after neoadjuvant or adjuvant treatments, is there an indication to propose it to patients in the same measure. Excellent iconography, English needs improvement. The bibliography supports the initial thesis.
Our response. Thank you for your valuable comments concerning our systematic review. In accordance with your proposal, we edited the English language used in this manuscript and made several changes in all parts of the manuscript. Regarding your question of combined treatment for the main lesion together with secondary hepatic lesions, the following text was added in the Discussion section: “In patients with synchronous colorectal liver … and should be evaluated as standard.”
Thank you for considering our revised manuscript.